# Veterans Health Administration staff experiences with suicidal ideation screening and risk assessment in the context of COVID-19

**Summer Newell** [1]*, **Lauren Denneson**[1,2], **Annabelle Rynerson**[1], **Sarah Rabin**[1], **Victoria Elliott**[1], **Nazanin Bahraini**[3,4], **Edward P. Post**[5,6], **Steven K. Dobscha**[1,2]

1 Center to Improve Veteran Involvement in Care (CIVIC), VA Portland Health Care System, Portland, OR, United States of America, 2 Department of Psychiatry, Oregon Health & Sciences University, Portland, OR, United States of America, 3 Rocky Mountain Mental Illness Research, Education, and Clinical Center, Aurora, Colorado, United States of America, 4 University of Colorado School of Medicine, Aurora, Colorado, United States of America, 5 VA Ann Arbor, Center for Clinical Management Research, Ann Arbor, MI, United States of America, 6 Department of Medicine, University of Michigan Medical School, Ann Arbor, MI, United States of America

* Summer.Newell@va.gov

**Data Availability Statement:** We as authors are fully supportive of sharing our study data. Per current Veterans Health Administration (VHA)

## Abstract

Universal screening for suicidal ideation in primary care and mental health settings has become a key prevention tool in many healthcare systems, including the Veterans Healthcare Administration (VHA). In response to the coronavirus pandemic, healthcare providers faced a number of challenges, including how to quickly adapt screening practices. The objective of this analyses was to learn staff perspectives on how the pandemic impacted suicide risk screening in primary care and mental health settings. Forty semi-structured interviews were conducted with primary care and mental health staff between April-September 2020 across 12 VHA facilities. A multi-disciplinary team employed a qualitative thematic analysis using a hybrid inductive/deductive approach. Staff reported multiple concerns for patients during the crisis, especially regarding vulnerable populations at risk for social isolation. Lack of clear protocols at some sites on how to serve patients screening positive for suicidal ideation created confusion for staff and led some sites to temporarily stop screening. Sites had varying degrees of adaptability to virtual based care, with the biggest challenge being completion of warm hand-offs to mental health specialists. Unanticipated opportunities that emerged during this time included increased ability of patients and staff to conduct virtual care, which is expected to continue benefit post-pandemic.

## Introduction

Screening for suicidal ideation and assessing patients for suicide risk has become a key component of suicide prevention efforts in healthcare settings, including the Veterans Health Administration (VHA) [1,2]. The joint VA-Department of Defense Clinical

guidelines, data from this study, which consist of confidential interviews with VHA employees, may be accessed by request only. Requests for data access will be considered and responded to within one month of the request and, subject to VHA regulation, de-identified datasets will be made available electronically. Requests must be made in writing to the Director of the Center to Improve Veteran Involvement in Care (CIVIC) Data Repository, Dr. Steven Dobscha (steven. dobscha@va.gov).

**Funding:** This work was supported by funding awarded to SD by the United States Department of Veterans Affairs (VA), Veterans Health Administration, VA Health Services Research and Development (IIR 19-215). https://www.hsrd. research.va.gov/funding/ The views expressed in this article are those of the authors and do not necessarily reflect the position or policy of the Department of Veterans Affairs or United States government. The funders had no role in study design, data collection and analysis, decision to publish, or preparation of the manuscript.

**Competing interests:** The authors have declared that no competing interests exist.

Practice Guidelines for the Assessment and Management of Patients at Risk for Suicide [3], published in 2019, recommends universal screening using a validated screening tool to identify individuals at risk and comprehensive follow-up assessment of individuals identified as being at risk. In accordance with these guidelines, the VHA recently implemented a multi-stage suicide risk assessment protocol across primary care, emergency department, and specialty mental health settings called VA Suicide Risk Identification Strategy, or "Risk ID" [4]. In early 2020, we initiated a study to understand Veteran and staff perceptions of Risk ID and how screening processes and perceptions of screening may impact subsequent care.

The coronavirus pandemic that emerged in March of 2020 [5] necessitated shifting healthcare visits to telephone or video platforms rapidly, pushing clinicians and other staff to quickly scale up telehealth and video technologies, policies, and procedures. This shift to virtual care has strong potential to affect risk screening and assessment processes due to challenges to establishing rapport in video- or telephone-meditated visits, unanticipated technological failures, or difficulties coordinating care with other staff and resources. Prior studies have suggested that risk assessment or screening should take place in the context of a trusting relationship with a provider, and that providers may often rely on colleagues for assistance [6–8]. Increasing physical and psychological distance between patients and their clinicians through telephone or video visits may make developing trust more challenging and further test clinicians' feelings of uncertainty about conducting risk assessment–ultimately hindering effective response to patients' clinical needs.

During the early phase of the pandemic, our study team began interviewing clinicians and nursing staff from primary care, mental health, and emergency department settings about their experiences conducting screening and risk assessment for suicide using VA's Risk ID procedures. To better understand how the pandemic impacted screening and risk assessment, we also asked staff participants directly about impacts of the pandemic on these processes. In this manuscript, we report on these findings, and discuss clinical implications for screening and suicide risk assessment in the context of a sudden shift to virtual care.

## Materials and methods

At the time of the interviews, the Risk ID process consisted of three stages: question 9 of the Patient Health Questionnaire-9 (PHQ-9), the Columbia Suicide Severity Rating Scale Screener (C-SSRS) [9], and VA Comprehensive Suicide Risk Evaluation (CSRE), a structured clinical assessment tool developed internally by the VA which inquires about factors critical to suicide risk [4]. Patients who screened positive on the PHQ-9 question (response of 'yes' to the question "thoughts that you would be better off dead or of hurting yourself in some way") were screened using the C-SSRS; a positive C-SSRS (defined as yes response to items 3, 4, 5, or 6b.) led to same-day completion of the CSRE [4]. Over the past few decades, VHA has implemented the Primary Care-Mental Health Integration (PC-MHI) initiative which places specially trained mental health specialists on-site to support primary care teams [10]; PC-MHI staff are often called in to assist with risk assessment following positive C-SSRS screens.

Data were collected from primary care and mental health staff at 12 VHA facilities across the U.S. between April and September 2020. Potential facilities were purposively identified from the larger pool of 171 VHA facilities nationally to reflect a range of characteristics including regional and geographic variability, operative complexity level, size (patient capacity), and adherence to the Risk-ID initiative (based on several performance measures). Facility directors were then contacted via email to invite them to participate. After receiving

facility leadership permission, the study team contacted primary care, mental health, and emergency department leads to disseminate recruitment emails to their staff. Those interested in participating contacted our project coordinator to schedule interviews. Forty participants completed interviews. Eighty-three percent of participants identified as women (n = 33), 68% identified as white and non-Hispanic (n = 27), 15% identified as African American (n = 6), 8% identified as Asian American (n = 3), 5% identified as white and Latinx (n = 2), and 5% identified as white and Middle Eastern (n = 2). Participants included: Six physicians, seven nurse practitioners, six psychologists, five licensed clinical social workers, one physician's assistant, one psychiatrist, eight registered nurses, three licensed practical nurses, one advanced medical support assistant, one peer support specialist, and one program manager. The average length of time participants had worked for the VA was seven years (range 2 months to 32 years), and the average time elapsed since training was 14 years (range 1 to 41 years).

The study team developed a semi-structured interview guide informed by our overall study research questions (S1 Appendix). COVID-19 was declared a pandemic by the World Health Organization (WHO) [5] shortly before recruitment began. In response, the authors adapted the interview guide to include questions regarding changes to the screening process as a result of COVID-19 and the shift to virtual care.

All interviews were audio-recorded and transcribed. Transcriptions were analyzed using Atlas.ti software by four coders: two primary coders and two secondary coders. The interdisciplinary coding team consisted of two research assistants and two experienced qualitative researchers including a sociologist and a social psychologist. The coding team met weekly to discuss emerging themes and new code categories and consulted co-authors with expertise in psychiatry and clinical psychology during analyses. We implemented an inductive-deductive hybrid approach for thematic analysis. Using the interview guide and research questions, the authors created an initial codebook. Each author independently reviewed three transcripts using the initial codebook, followed by a meeting to discuss and refine the first iteration of the codebook. Codes were added or amended during the coding process to capture themes not previously defined. Data used for the current analysis were limited to specific discussions of the COVID-19 pandemic and care and screening changes due to the pandemic. All authors discussed themes arising from these data until agreement was reached on main findings.

This study was reviewed and approved by the joint institutional review board (IRB) of the medical center and university at which the study was jointly conducted. A waiver of written informed consent was approved; all participants provided verbal consent to be recorded prior to interviews.

## Results

At the time of interviews, participant sites were at various stages of adapting to social distancing and technological requirements brought on by the pandemic. This enabled us to learn about a range of experiences regarding challenges and concerns, adaptations, and potential opportunities arising from pandemic-related changes. Some sites were able to pivot to telephone or video-based care screenings quickly, whereas others struggled. All sites reported adaptations they made in screening processes to continue to assess and meet patient needs, as well as unexpected opportunities that arose that could continue following the pandemic. There were overlaps in the themes elicited by primary care and mental health staff, but each setting also generated unique themes.

## Increased concerns for patient well-being

Consistent among all sites and both care settings was a concern for patients who were experiencing increased social isolation.

> I think there is extreme loneliness. And I'm not sure how we combat that, but then given this COVID stuff where everybody is staying inside, that's most of the comments that I've been getting on the telephone appointments. I've been trying to ask them, how are you doing amidst this COVID shutdown? What's happening with you? And most of them are saying I'm doing pretty well, although I'm getting bored. You know? But you can see where, I mean, even young people have so much loneliness. Like, I had a guy who, his dreams didn't come true in the Navy, and he lives by himself. He doesn't have anybody to talk with. He doesn't have friends. He doesn't interact with his family. And I think, oh my God. This guy is so much at risk. *Nurse Practitioner, PC Setting*

One factor leading to this concern was cancellation of all in-person mental health and other health related groups that offer social interaction, and some staff speculated that this may lead to increased suicidal ideation for some patients.

> I think before, we had a lot of great resources at [site]. I mean, there's a lot of groups going on. PTSD groups. A lot of different groups for all sorts of Veterans. And the ones that were in it, liked it. They had tai chi. They had yoga; they had all that. Post-COVID, they're having a hard time. *Nurse Practitioner, PC setting*

Vulnerable patients, including those needing to enroll in inpatient substance use treatment or experiencing homelessness had less access to resources to meet their needs.

> Some of them, before—pre-COVID, a lot of them were already in the shelter, but post—I mean, currently, with COVID, with a lot of shelters that's closed down or not accepting that many people without jumping through hoops to get there, COVID screening or just the social distancing, it's a little bit harder, so we're seeing more people answer yes to suicidal ideations. *Physician's Assistant, MH setting*

Participants reported that for some patients, the inability to physically come to the office further exacerbated their social isolation.

> I think for some patients, it's about the experience. You're leaving your house. These are some patients who maybe live by themselves, and leaving their houses, [sic] they come to the clinic, they make themselves a cup of coffee, they get a snack because the volunteers always bring snacks. Then they come into my office and they talk to me. Now it's not the same because they're just sitting in their house doing that and they're not having that experience and I think that's what they miss. I don't think it's seeing my face or missing me. I'm still the same person, I'm just through a video on their phone. *LCSW*

## Screening patients during pandemic challenged quality of care

Both primary care and mental health settings increased use of virtual care for their patients, and the biggest challenge regarding screening reported by primary care staff during this time was concern with the ability to conduct warm handoffs to mental health providers in a timely

manner if additional assessment or follow-up by a specialist was required. Staff had to rely on instant messaging to find support for the patients, but this was not always efficient.

> . . .this has been a learning experience. It's been kind of a mess, you know, like they'll answer my call, five minutes later, the doctor will call, they won't answer. And it's just back and forth, back and forth. . .That's the hardest part. *LPN*

Ensuring the safety of a patient reporting having suicidal thoughts via phone or video was a concern among both primary care and mental health staff.

> For me I think if I've got somebody over the phone and there's clear concern for suicide, I would want to make sure that they're safe first. You know if they're mentioning that they're having thoughts of suicide, you know are you someplace safe, have you made a plan? As far as do you have the gun in your hand, because that's going to be the focus of safety. Where if somebody is sitting in my office, I can tell that they're safe right in front of me. You know? *LCSW*

Some staff reported that screenings were not being conducted at all because clear protocols did not exist for what to do when a patient screens positive for SI.

> So a lot of people are just not asking them at all. . .A lot of people are talking about the liability and not being able to do the warm handoff. *Physician*

Staff members were concerned about how the virtual formats might feel for some patients —and how that might limit their ability to be forthcoming about their mental health experiences.

> I think the technology, whether it's VVC or just on the telephone, it's just so impersonal. I just feel that way. *LPN*

> Because and now with COVID, there's a lot of situations where people don't feel comfortable asking these questions over the phone or over video and Veterans don't feel comfortable because they feel they're being recorded. *Psychologist*

Mental health staff reported being more comfortable screening for SI via video or telephone compared to primary care but reported that it is more difficult without body language and other cues to complete a full assessment.

> I'd like to have eyes on him so I can make an assessment because I know, it's hard to explain, when you're doing mental health, you have to make a diagnosis. That diagnosis is based off a lot of different things. It's based off of how they present, what they're talking about with all of these different components of that and if you can't see this person, you can't really see what their affect is. You can't really see what their face is like. It does make it more difficult. *LCSW*

This lack of non-verbal cues was especially difficult when making decisions about whether to hospitalize patients who were experiencing SI.

> So I was apprehensive when this whole thing started, because especially for patients who might need to be 5150'ed [term used for involuntary hospitalization] there's less of a control, should I say? They're not right there with you. *Psychiatric Nurse Practitioner*

## Adapted screening processes for virtual care

Despite the reported challenges involved with pivoting to virtual care, most sites were able to adapt to the circumstances to ensure patients continued to be screened. To reduce the chances of a patient "falling through the cracks" during a handoff, some sites adapted their screening systems to solely have licensed providers conduct screenings, rather than the nurse or other non-physician staff, reducing some of the anxiety reported by these staff when they were unable to contact the needed specialist when a patient screened positive.

> Well, we actually haven't been doing them if we are teleworking, because if they screen positive, and my doctor—because I'll call and do their check-in questions, like I normally would in the clinic, prior to their appointment, that way they're ready to go right at their appointment time. But if they screen positive, I can't give them a warm hand-off to my doctor. Because I can't transfer phones from home, so we don't do those and the doctors have been doing those screenings. *LPN*

Some participants were selective in which patients they screened and reduced the formality of the screening to allow for flexibility in how to respond.

> Yeah, I think like, just basically informal things. Like, how are you feeling and what's going on? And if they are saying that there is anything of concern, then we're prompting—Do they feel safe? Do they carry a gun—you know, we'll ask those questions, but we're not like, screening or going off a checklist about that, specifically. It's almost like, we talk about it because something comes up. But we're not asking every person when we talk to them. *RN*

Other participants informally added some questions about recent losses related to the pandemic to make sure they were capturing recent events.

Mental health staff reported fewer adaptations to virtual screenings processes although many reported asking patients 1) where were they located; 2) were they alone, and if not, who was in the home; and 3) do they have sufficient privacy. Others increased their communication with the patient about the purpose and process of screening and assessment.

> I mean I kind of, everything I do in person is what I do. And I just kind of acknowledge this is over telephone or VVC, when I'm reading off of self-report measures I try to lighten the mood and just say, I might sound a little bit like a robot because I'm reading verbatim these items, so just bear with me. And so I try to be really good about reading the instructions for each self-report measure, their answer options, and then each item as it appears. And so it's been working well. *Psychologist*

## Unexpected benefits of pivoting to virtual care

Several participants reported unexpected opportunities for improved care that arose during the pandemic. Many participants, but especially mental health care staff, lauded the increased capacity provided by VHA to conduct virtual care, particularly video-based care.

> We didn't have a lot of people prior to COVID be set up for VVC appointments. And now with this pandemic or whatever, more and more people are getting approved for it, obviously. Or they'll qualify for a VA issued iPad because they're seen frequently enough to meet the criteria. . .So, you know, I do think they're more apt to follow up that way. You

know? So, I mean I do think that it's some good that has come out if this is I do think that more people have been attending their appointments. *RN, MH setting*

Several staff reported that the ability to participate in virtual care was a pleasant surprise for some patients—especially given the many barriers that come with in-person care such as transportation and scheduling.

I think people actually are really enjoying virtual, actually. . .Maybe because our VA is really hard to get to. . .You know, traffic is awful, parking is awful. *LPN*

Although increased use of technology by both patients and staff was considered a remarkable improvement, several participants cautioned that this was not a solution for all patients, particularly older patients who are less inclined to use technology. These patients were able to meet by telephone, but this was less ideal than video-based care, and even less ideal than in-person visits.

Participants reported that more appointments are being kept, and this was especially true for mental health visits.

People are definitely keeping more appointments in mental health in VVC and telephone because they don't have the obstacles of also coming to the clinic, waiting in line, checking in. They don't have to deal with that. They can literally just from the home computer, just turn it on and their appointment's done, they sit down as soon as it starts and they're at home already when it's over. I think that they really do like that and I think we're getting better turn out because of that. *RN, MH setting*

Despite the increased rate of patients keeping appointments, participants reported that fewer appointments were being scheduled overall, especially in primary care. This reduction in appointments was thought to be due to the pandemic. While this caused concern because patients are receiving less care, the lower census of patients allowed for staff to provide increased follow-up to their at-risk patients experiencing social isolation and other concerns.

I feel like if anything, with our lower volumes, we're able to follow up more closely with patients that we're worried about. And there's certainly patients that we're picking up based on CAN scores [a measure to assess risk of hospitalization or mortality], that score high, based on hospitalizations and ER visits, that are also sort of high-risk suicide flagged. *Physician researcher*

## Discussion

To our knowledge, this is the first analysis to explore impacts of a pandemic on universal screening processes for suicide risk from the perspective of clinicians and staff. This analysis highlighted challenges and concerns as well as adaptations in processes and indirect benefits that have potential to impact care after the pandemic subsides.

A frequent theme among participants was concern about their patients' welfare. Staff were concerned about impacts of social isolation in general but also potential disruptions in care among those individuals who may depend on the healthcare system for social connection and support. Rapid elimination of group offerings may be especially impactful for this subgroup of individuals, and it could be important to further develop plans and technology platforms to

better equip patients and clinical staff to be able to shift to, and sustain, group treatments during crises that limit in-person gatherings.

An additional key concern of participants regarded challenges in reliably assessing patients' level of risk. In particular, staff were concerned about the inability to see other potential clues pertinent to determining level of risk such as body language and were afraid they would miss something that would indicate greater attention was needed. Concern was expressed about virtual technology limiting patients' disclosure of sensitive information, which might have been compounded among older patients who some felt were less inclined to use technology. These findings suggest that health and other systems may wish to invest more in training people to use healthcare-related technology and developing more user-friendly platforms for vulnerable individuals. The data also bring to light the importance of adequate security and privacy protections for telehealth systems to increase patient trust and comfort in disclosing sensitive information through a virtual platform. One benefit of adaptations made for COVID-19 is that many individuals (patients and staff) who were previously uncomfortable virtual care have gained knowledge and experience which may help to lessen this impact going forward.

Ensuring the safety of participants should they have positive screens or other concerning behaviors was a consistent theme for staff, citing lack of or delay in development of clear procedures to follow compounded concerns. In usual practice, if a nurse conducts a screening and it is positive, the primary care provider is readily available to carry on with further assessment. Further, in recent years, PC-MHI programs have dramatically enhanced warm-handoff capability; that is, being able to connect patients felt to be at risk in primary care immediately with mental health specialists. Although it is possible to make these handoffs virtually, it can be cumbersome, especially if a patient is at higher risk and thus potentially requires hospitalization. Sites varied in their approaches in response to this set of challenges, with some nurses deferring initial screening to licensed providers or other screeners ensuring they had good contact information for the patient in case emergency services were required. Looking forward, development and dissemination of clear procedures and expectations for screening in a virtual environment and following up on screens is needed.

In addition to patients and clinicians gaining new knowledge and experience working through virtual platforms, other adaptations and perhaps unexpected positive impacts were noted. Patients who were more stable or for whom the care team was less concerned about had decreased health care appointments due to the pandemic, and this provided opportunity for the care team to prioritize and reach out to known higher-risk patients. We also found that some facilities were able to shift their processes rapidly to accommodate the new circumstances, especially those who already had some virtual care in place. Others struggled considerably and further analyses of reasons for this variation has potential to yield important insights about factors influencing system-level crisis response.

There are several limitations worth noting. This was a qualitative study—the project was designed to identify key themes and generate hypotheses for further study, and we purposively interviewed individuals across a broad range of sites and disciplines. Despite having variation in site characteristics and staff roles, our data did not reveal sufficient contextual differences between sites to explain the varying levels of adaptability. As such, contextual factors which might allow for examination and comparisons of facility-level approaches are generally absent. Although participants were from various facilities across the country, the sample included only staff who served Veterans within the context of the VHA. We also did not include Veteran perspectives in this analysis; staff reports of patient impacts and other issues may be misattributed in some cases. Finally, the interviews were done rather early during the COVID-19 pandemic. Care processes, policy, and technological options have continued to evolve, and our results may have less applicability to current practice than they did a year ago.

There are several lessons to be learned from our findings. First, overall, staff adapted to the circumstances and maintained flexibility as they attempted to cope with rapidly changing circumstances—their dedication to care and to Veterans was evident. Although systems have continued to evolve since COVID-19 began, our results suggest that additional preparation should be made for future pandemics or other disaster situations that decrease physical access. These preparations include continued development, implementation, and perhaps even practice using virtual platforms by clinicians and patients, as well as improving broadband access, especially for more vulnerable individuals who rely heavily on healthcare systems for social connection and support. Such preparations can help ensure that rapid shifts to virtual care increase access to mental health services during and post-pandemic [11] without exacerbating health disparities. Health systems would also benefit from pro-active development of policy and procedures to enact should in-person access become suddenly unavailable. For example, individual facilities might wish to conduct rigorous needs assessments or failure mode (and effects) analysis exercises to better prepare for rapid shifts to virtual care during times of crisis.

## Supporting information

**S1 Appendix. Interview guide: Primary care staff.**
(DOCX)

## Acknowledgments

The authors acknowledge and appreciate the efforts made by the Health Services Research and Development (HSR&D) Centralized Transcription Service Program (CTSP) to complete transcription of all interviews for this project.

## Author Contributions

**Conceptualization:** Lauren Denneson, Nazanin Bahraini, Edward P. Post, Steven K. Dobscha.

**Formal analysis:** Summer Newell, Lauren Denneson, Annabelle Rynerson, Sarah Rabin.

**Funding acquisition:** Steven K. Dobscha.

**Investigation:** Summer Newell.

**Methodology:** Summer Newell, Lauren Denneson.

**Project administration:** Annabelle Rynerson, Sarah Rabin, Victoria Elliott.

**Supervision:** Summer Newell, Lauren Denneson, Victoria Elliott, Steven K. Dobscha.

**Writing – original draft:** Summer Newell, Lauren Denneson, Annabelle Rynerson, Sarah Rabin, Steven K. Dobscha.

**Writing – review & editing:** Summer Newell, Lauren Denneson, Annabelle Rynerson, Sarah Rabin, Victoria Elliott, Nazanin Bahraini, Edward P. Post, Steven K. Dobscha.

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
