## [Decision Letter · Decision Letter 0]

1 Oct 2021

PONE-D-21-22480Veterans Health Administration staff experiences with suicidal ideation screening and risk assessment in the context of COVID-19PLOS ONE

Dear Dr. Newell,

Thank you for submitting your manuscript to PLOS ONE. After careful consideration, we feel that it has merit but does not fully meet PLOS ONE’s publication criteria as it currently stands. Therefore, we invite you to submit a revised version of the manuscript that addresses the points raised during the review process.

We look forward to receiving your revised manuscript.

Kind regards,

Sarah A. Arias

Academic Editor

PLOS ONE

Journal Requirements:

Reviewers' comments:

Reviewer's Responses to Questions

**Comments to the Author**

1. Is the manuscript technically sound, and do the data support the conclusions?

Reviewer #1: Yes

2. Has the statistical analysis been performed appropriately and rigorously? 

Reviewer #1: N/A

3. Have the authors made all data underlying the findings in their manuscript fully available?

Reviewer #1: Yes

4. Is the manuscript presented in an intelligible fashion and written in standard English?

Reviewer #1: Yes

5. Review Comments to the Author

Reviewer #1: Overall, this was a well-written and thoughtful manuscript reporting clinical impressions of the VA transition to remote administration of Risk ID in the early stages of the COVID crisis. My primary concerns are: 1) that some of the background information about VA practices is not included and is not easily accessible to readers outside of that system, and 2) use of neutral language. Please see comments below:

Materials and Methods:

The opening paragraph of the methods refers to two VA initiatives (VA CSRE; PC-MHI) without providing citations or links to any online documentation. This information should be easily accessible to non-VA affiliated readers who may not be able to properly contextualize the methods and results without supplementary information.

Results

Language should be more neutral in places. Phrases like “deep concern” instead of just “concern” would be better. Though I agree with the sentiment, this statement from the Discussion read as a value judgement:

"First, overall, staff exhibited ingenuity and flexibility as they attempted to cope with rapidly changing circumstances—their clear dedication to care and to Veterans was evident."

Discussion

The diversity of expertise among participants and hospital size and setting was a strength of the paper, but greater consideration of how those differences aligned with themes is warranted. For example, there is some discussion at length about a shift away from having nurses administer Risk ID. Was this occurring at larger hospitals, rural hospitals, etc.? Though COVID has been challenging across levels of expertise, did baseline staff experience impact decisions to shift assessment assignment?

The discussion of changes in quality of care would be more useful if the results were contextualized in the hospital setting, e.g., were there particular types of hospitals, or staff makeup, that were more typical of hospitals that made the decision to not carry out parts or Risk ID?

I agree with the passage below, but what kinds of evaluations do the authors recommend? As a closed health care system, there are some unique opportunities for addressing such questions at a depth that is more difficult to achieve when working in a majority private health insurance setting. It would be good if the authors made some more concrete recommendations.

"Others struggled considerably and further analyses of reasons for this variation has potential to yield important insights about factors influencing system-level crisis response."

In the discussion, the authors draw distinctions between hospitals who were, and were not, able to transition to virtual care, but the discussion of characteristics of both groups was shallow. I appreciate that this is a preliminary, hypothesis-generating piece, but I don't have a strong sense of where the next logical step in this line of research lies. How would qualitative work like yours be integrated into larger scale program evaluation efforts?

6. PLOS authors have the option to publish the peer review history of their article (what does this mean?). If published, this will include your full peer review and any attached files.

Reviewer #1: No

---

## [Author Response · Author response to Decision Letter 0]

9 Nov 2021

Reviewer #1: Overall, this was a well-written and thoughtful manuscript reporting clinical impressions of the VA transition to remote administration of Risk ID in the early stages of the COVID crisis. My primary concerns are: 1) that some of the background information about VA practices is not included and is not easily accessible to readers outside of that system, and 2) use of neutral language. Please see comments below:

Author response: We will respond to specific comments where they occur below. 

Materials and Methods:

The opening paragraph of the methods refers to two VA initiatives (VA CSRE; PC-MHI) without providing citations or links to any online documentation. This information should be easily accessible to non-VA affiliated readers who may not be able to properly contextualize the methods and results without supplementary information.

Author response: Thank you for pointing out this oversight. We have provided additional description and added a relevant citation. 

Results

Language should be more neutral in places. Phrases like “deep concern” instead of just “concern” would be better. Though I agree with the sentiment, this statement from the Discussion read as a value judgement:

"First, overall, staff exhibited ingenuity and flexibility as they attempted to cope with rapidly changing circumstances—their clear dedication to care and to Veterans was evident."

Author response: We agree that some of the language holds some value judgement. We have edited the language to be more neutral. 

Discussion

The diversity of expertise among participants and hospital size and setting was a strength of the paper, but greater consideration of how those differences aligned with themes is warranted. For example, there is some discussion at length about a shift away from having nurses administer Risk ID. Was this occurring at larger hospitals, rural hospitals, etc.? Though COVID has been challenging across levels of expertise, did baseline staff experience impact decisions to shift assessment assignment?

The discussion of changes in quality of care would be more useful if the results were contextualized in the hospital setting, e.g., were there particular types of hospitals, or staff makeup, that were more typical of hospitals that made the decision to not carry out parts or Risk ID?

Author Response: Thank you for the suggestion to better describe the contextual differences in the focal clinics. The study was not designed for, and our qualitative sample did not allow for, meaningful comparison between sites. We intentionally sought out variation across these characteristics to ensure broad representation of ideas and experiences, which can lead to better identification of key themes and hypotheses for later study (Sofaer, 1999 p. 1104). We did reexamine our data, and as expected, due to substantial variation across site and provider characteristics we were unable to detect clear patterns of association. We do note in the discussion (line 367) that sites that had existing virtual care prior to the pandemic might have had an easier time adjusting. Further, we highlight the limited contextual data as a limitation. 

I agree with the passage below, but what kinds of evaluations do the authors recommend? As a closed health care system, there are some unique opportunities for addressing such questions at a depth that is more difficult to achieve when working in a majority private health insurance setting. It would be good if the authors made some more concrete recommendations.

"Others struggled considerably and further analyses of reasons for this variation has potential to yield important insights about factors influencing system-level crisis response."

In the discussion, the authors draw distinctions between hospitals who were, and were not, able to transition to virtual care, but the discussion of characteristics of both groups was shallow. I appreciate that this is a preliminary, hypothesis-generating piece, but I don't have a strong sense of where the next logical step in this line of research lies. How would qualitative work like yours be integrated into larger scale program evaluation efforts?

Author Response: We agree that we did not outline clear next steps for further work. Given that we did not have sufficient data to identify contextual differences between sites that adapted their SI protocols versus those that did not, further work that more precisely measures contextual variables is suggested. We have outlined further recommendations on line 396. 

Reference

Sofaer, S. (1999). Qualitative methods: what are they and why use them?. Health services research, 34(5 Pt 2), 1101.

---

## [Decision Letter · Decision Letter 1]

14 Dec 2021

Veterans Health Administration staff experiences with suicidal ideation screening and risk assessment in the context of COVID-19

PONE-D-21-22480R1

Dear Dr. Newell,

We’re pleased to inform you that your manuscript has been judged scientifically suitable for publication and will be formally accepted for publication once it meets all outstanding technical requirements.

Kind regards,

Sarah A. Arias

Academic Editor

PLOS ONE

Reviewers' comments:

Reviewer's Responses to Questions

**Comments to the Author**

1. If the authors have adequately addressed your comments raised in a previous round of review and you feel that this manuscript is now acceptable for publication, you may indicate that here to bypass the “Comments to the Author” section, enter your conflict of interest statement in the “Confidential to Editor” section, and submit your "Accept" recommendation.

Reviewer #1: All comments have been addressed

2. Is the manuscript technically sound, and do the data support the conclusions?

Reviewer #1: Yes

3. Has the statistical analysis been performed appropriately and rigorously? 

Reviewer #1: Yes

4. Have the authors made all data underlying the findings in their manuscript fully available?

Reviewer #1: Yes

5. Is the manuscript presented in an intelligible fashion and written in standard English?

Reviewer #1: Yes

6. Review Comments to the Author

Reviewer #1: (No Response)

7. PLOS authors have the option to publish the peer review history of their article (what does this mean?). If published, this will include your full peer review and any attached files.

Reviewer #1: No

---

## [Editor Report · Acceptance letter]

17 Dec 2021

PONE-D-21-22480R1 

Veterans Health Administration staff experiences with suicidal ideation screening and risk assessment in the context of COVID-19 

Dear Dr. Newell:

I'm pleased to inform you that your manuscript has been deemed suitable for publication in PLOS ONE. Congratulations! Your manuscript is now with our production department. 

Kind regards, 

on behalf of

Dr. Sarah A. Arias 

Academic Editor

PLOS ONE